

# Increasing leaf sizes of the vine *Epipremnum aureum* (Araceae): photosynthesis and respiration

Carolina Brito[1], Dulce Mantuano[2], Karen L.G. De Toni[3] and André Mantovani[3]

[1] Instituto de Pesquisas Jardim Botânico do Rio de Janeiro, Escola Nacional de Botânica Tropical, Programa de Pós-Graduação em Botânica, Rio de Janeiro, Rio de Janeiro, Brazil

[2] Departamento de Botânica, Universidade Federal do Rio de Janeiro, Rio de Janeiro, Rio de Janeiro, Brazil

[3] Diretoria de Pesquisas, Instituto de Pesquisas Jardim Botânico do Rio de Janeiro, Rio de Janeiro, Rio de Janeiro, Brazil

Corresponding author
André Mantovani, andre@jbrj.gov.br

## ABSTRACT

The canopy leaves of allomorphic aroid vines can exceed 2,000 cm$^2$, up to 30 times larger than respective understorey leaves. In the literature, this allomorphic increase in leaf area of aroid vines was hypothesized to improve its light foraging capacity. The viability of these large leaves depends on carbon acquisition obtained from their larger area and on the respective costs of production, maintenance and support. To evaluate and understand how leaf enlargement affects performance, we analyzed the photosynthesis and respiration of *Epipremnum aureum* leaves of different sizes via photosynthetic response light curves, morpho-physiology and anatomical parameters. Leaf size was increased by varying growth direction (horizontal *vs.* vertical) and light conditions (low *vs.* high). Vertical plants in high light produced leaves 9–13 times larger than those under other conditions. Saturated photosynthetic rates per area were similar across leaves of *E. aureum*, regardless of size, but respiration rates increased while specific leaf area decreased in larger leaves. This may suggests that larger leaves do not offset their costs per unit area in the short term, despite field observations of continuous enlargement with increased plant size. However, the high light levels able to saturate photosynthesis under field conditions are achieved only by larger leaves of *E. aureum* positioned at canopies (PPFD around 1,000 $\mu$mol m$^{-2}$ s$^{-1}$), not occurring at understory where smaller leaves are positioned (PPFD around 100 $\mu$mol m$^{-2}$ s$^{-1}$). This is confirmed by the higher values of the relative growth rate (RGR) and net assimilation rate (NAR) parameters exhibited by the vertical plants in high light. The saturated photosynthetic rates found here under experimental conditions for the smaller leaves of *E. aureum* could be related to their high invasive capacities as alien species around the world. We propose that the costs of larger aroid leaves might be outweighed by a strategy that optimizes size, morphophysiology, anatomy, photosynthesis and, lifespan to maximize lifetime carbon gain in tropical forests.

## INTRODUCTION

Aroid vines, also referred to as nomadic vines or lianescent aroids, are a group of tropical species in the Araceae family that are ubiquitous in humid tropical forests (*Moffett, 2000*; *Caleño Ruíz, Rodríguez-Eraso & López-Camacho, 2018*; *Filartiga et al., 2021*; *Zotz et al., 2021*). In the early stages of development, these vines grow plagiotropically, *i.e.,* they grow horizontally on the ground until they find a bearing host. At this point, they switch to orthotropic growth and grow vertically towards the canopy (*Strong & Ray, 1975*). Along with the transition from plagiotropic to orthotropic growth, the aroids undergo a morphogenesis that can lead to a size increase and a different morphology of the leaves produced in the canopy compared to those produced on the forest floor (*Ray, 1990*). The synergistic effect of contact (with the phorophyte), vertical growth and light exposure is the main morphogenetic trigger inducing leaves of aroid vines to increase 20–30 times in comparison to leaves growing near the ground (*Steinitz & Hagiladi, 1987*; *Mantuano et al., 2021*; *Brito et al., 2022*). Not only do leaf area changes occur during this morphogenetic process but also changes to petiole, stem and aerial root dry mass (*Brito et al., 2022*) and morphophysiologies (*Filartiga, Vieira & Mantovani, 2014*; *Mantovani, Mantuano & De Mattos, 2017*). Physiological and anatomical adaptations occur in large aroid leaves produced in the canopy, such as higher electron transport rates (ETR), higher chlorophyll a/b ratio and larger intercellular spaces (*Mantovani, Pereira & Mantuano, 2017*; *Mantovani, Brito & Mantuano, 2018*; *Mantuano et al., 2021*). These adjustments suggest an increase in light interception and photosynthetic capacities (*Ray, 1990*; *Brito et al., 2022*) when large leaves produced in the canopy substitute the small leaves produced at the ground level.

The leaf area of allo- and heteromorphic aroid vines increases under a more illuminated exposure and higher air evaporative demand of canopies (*Filartiga, Vieira & Mantovani, 2014*). The opposite response is generally found for terrestrial plants, whose leaf area is reduced under the same abiotic gradient, as it generates a positive effect on the water and energy balance of leaves (*Givnish & Vermeij, 1976*; *Givnish, 1988*). The question is how these bigger structures (large leaves, stems and roots) are produced and maintained against potential limitations to water and nutrient transport throughout the soil-to-canopy transition (*Domec et al., 2019*). These limitations are probably counterbalanced by anatomical, morphophysiological and photosynthetic adjustments that improve carbon and water relations (*Givnish & Vermeij, 1976*).

Although large leaves might improve light foraging (*Poorter & Rozendaal, 2008*), there must be compensation for the costs and risks of leaf construction and maintenance (*Milla & Reich, 2007*). In addition to the higher construction costs of a wider leaf, there is also a higher risk of desiccation, overheating, photoinhibition and herbivory (*Coley, 1983*; *Falster & Westoby, 2003*; *Wright et al., 2017*). These risks are usually associated with an increase in photo and dark respiration, which also reduces the overall carbon gain (*Raghavendra & Padmasree, 2003*; *Slot & Kitajima, 2015*; *Walker et al., 2016*). The higher costs of large leaves can be compensated for by improving their capacity for light interception and use, both in the short or long terms (*Wright et al., 2004*; *Niinemets, Portsmuth & Tobias, 2006*; *Poorter & Rozendaal, 2008*). This compensation occurs in the short or long term when net

carbon gain is modulated *via* lifetime to minimize the marginal costs of production and maintenance of leaves, stems and, roots (see Gmax theory in *Castorena et al., 2022*). Data in the literature have confirmed this idea in the short term but for just one species and without comparing photosynthesis and dark respiration simultaneously. Previous works showed that large canopy leaves of the aroid vine *Rhodospatha oblongata* (Araceae) presented higher photosynthetic rates compared to smaller ones produced in the understorey (*Mantuano et al., 2021*). In addition, the whole plant increases in size (*i.e.,* length, diameter and dry mass of stem and aerial roots) in this species while its large leaves are produced (*Filartiga, Vieira & Mantovani, 2014*), suggesting an improvement in plant growth. In the long term, an extended lifespan can improve investment made in large leaves by increasing its lifetime for carbon uptake (*Wright et al., 2004*; *Kikuzawa & Lechowicz, 2006*). The few available datasets in the literature indicate that aroid vines can present longer-lived leaves in comparison to trees and herbs in humid forests in Indonesia (*Shiodera, Rahajoe & Kohyama, 2008*).

This study investigated whether leaf enlargement of the aroid vine *Epipremnum aureum* could influence its performance and maintenance in the canopy. We hypothesized that large *E. aureum* leaves would present higher photosynthetic properties in comparison to smaller leaves, compensating for an expected higher respiration rate (*Cannell & Thornley, 2000*). These adjustments at the leaf level could increase carbon acquisition for the whole plant, helping to explain the size increase of aroid vines while climbing hosts in the forest.

## MATERIALS & METHODS

### Plant species and experimental setting

*Epipremnum aureum* is an aroid vine used as a model species in plant physiology experiments (*Khayyat, Nazari & Salehi, 2007*; *Di Benedetto, Galmarini & Tognetti, 2018*). This aroid species has a cosmopolitan distribution today, although its original occurrence was traced to the Solomon Islands, making it a potential invasive species in the Rio de Janeiro Atlantic Rain Forest (*Valadares, 2011*) as well as all around the world (*Hachisuka, Ishikawa & Ugawa, 2023*). In natural field conditions, this allomorphic aroid vine initially presents small leaves (around 50 cm$^2$) near the ground and larger leaves (around 2,000 cm$^2$) close to the canopy. The soil connection is maintained *via* aerial roots throughout the whole life cycle (*Mantovani, Pereira & Mantuano, 2017*).

To investigate the effects of the leaf area increase on the photosynthetic properties of *E. aureum*, we built an experiment following the procedures of *Steinitz, Hagiladi & Anav (1992)*. The experiment was conducted in a tropical open-air ventilated greenhouse at the Rio de Janeiro Botanical Garden, Brazil, over a period of 10 months, from July 2017 to April 2018. Two light conditions (low light and high light) and two growth directions (horizontal and vertical) were used as follows: low light–horizontal growth (LL-horiz), low light–vertical climbing (LL-climb), high light–horizontal growth (HL-horiz) and high light–vertical climbing (HL-climb) (Fig. 1). Individual rooted cuttings in the horizontal treatment were cultivated to creep along the substrate (*i.e.,* plagiotropic growth). For vertical treatments (*i.e.,* orthotropic growth), the cuttings were rooted in the substrate, and the stem nodes were fixed to a vertical support (bamboo sticks 15 cm in diameter). Plants

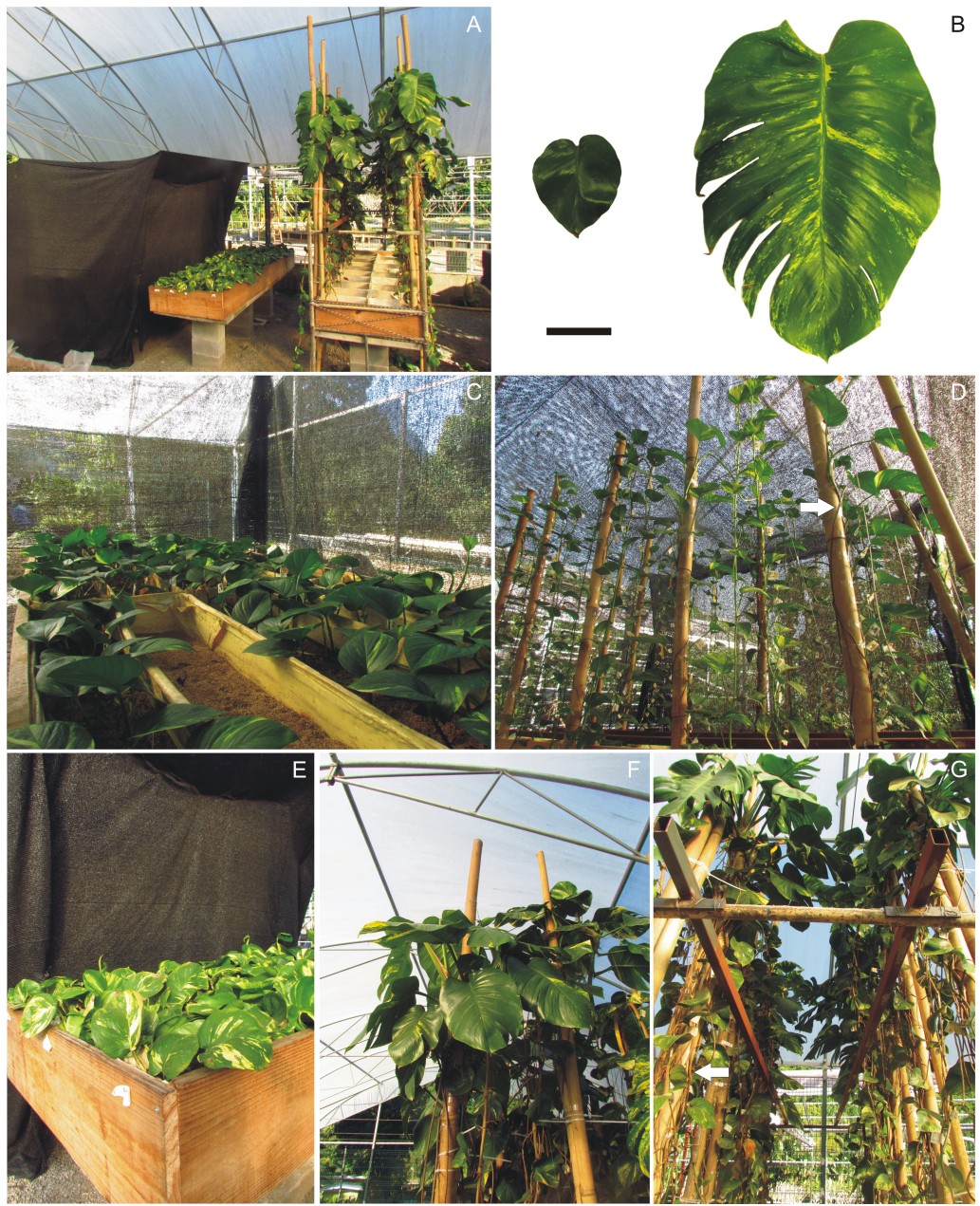

**Figure 1  Experimental setup for leaf photosynthesis analysis of *Epipremnum aureum* growing under different light conditions (low and high light) and growth directions (horizontal and vertical).**
(A) General view. From the left to right, black shade cloth pavilion (for the low light treatments 'LL-horiz' and 'LL-climb') and the two 'high light' treatments 'HL-horiz' and 'HL-climb'. (B) Small (LL-horiz) and large (HL-climb) leaves. Bar equals 10 cm. (C) LL-horiz individuals. (D) LL-climb individuals. Arrow indicate one aerial feeder root. (E) HL-horiz individuals. (F) HL-climb individuals. (G) HL-climb individuals from below. Arrow indicate many aerial feeder roots.

receiving the low-light treatment grew in a shade cloth pavilion. During the experimental period, the maximum photosynthetic photon flux density (PPFD) values recorded were 25.86 μmol photons $m^{-2}$ $s^{-1}$ for the low-light treatment and 622.25 μmol photons $m^{-2}$ $s^{-1}$ for the high-light treatment. Plants exposed to the low-light treatment received a maximum daily photon flux of 1.22 mol, while those under the high-light treatment received 29.70 mol of photons per day. The ratio of red to far-red light (R:Fr) remained consistent between both treatments, fluctuating between 0.8 and 1.2 throughout the day (*Brito et al., 2022*). For each experimental condition, 10 cuttings were used, totaling 40 individuals. For complementary information about experimental conditions, see *Brito et al. (2022)*.

**Photosynthetic parameters**

Light response curves were generated in April 2018 after 10 months of growth using gas exchange analysis and chlorophyll fluorescence measurements recorded using a Ciras II infrared gas analyzer (PP Systems, Amesbury, MA, USA) and a MINIPAM modulated fluorometer (H. Walz, Effeltrich, Germany), respectively. One completely expanded leaf obtained near the apex was selected from each of the five distinct individuals, determining five replicates per treatment. Both gas exchange and chlorophyll fluorescence analyses were simultaneously performed between 06:00 and 11:00 am. The air temperature and humidity inside the Ciras II chamber were 25 °C and 75% respectively, limiting any VPD effect. Soil watering was always higher (*i.e.,* less negative) than −0.7 MPa (*Brito et al., 2022*). After 5 min of leaf acclimation at 800 μmol $m^{-2}$ $s^{-1}$, gas exchange curves were obtained with 9 photosynthetic photon flux density (PPFD) values (1,800, 1,600, 1,200, 800, 500, 240, 120, 60, 0) at 380 PPM of $CO_2$. Measurements were obtained in each step after 10 min. From these curves, maximum carbon assimilation rate in area ($A_{max}$) obtained at light saturation point (LSP) and, dark respiration rates in area (Rd) obtained in the absence of light (by covering the leaf chamber with a dark cloth) were determined using a software macro routine from Microsoft Excel developed by *Lobo et al. (2013)* after adjusting light curves according to the equation proposed by *Prado & De Moraes (1997)*. The carbon assimilation rates at PPFD 100 μmol $m^{-2}$ $s^{-1}$ ($A_{100}$) for LL-horiz and at 1,000 μmol $m^{-2}$ $s^{-1}$ ($A_{1000}$) for HL-climb were also obtained from gas exchange curves and compared. These PPFD values respectively characterize the soil (LL-horiz) to canopy (HL-climb) transition of *E. aureum* in natural environments (*Mantovani, Pereira & Mantuano, 2017*).

Immediately after gas exchange measurements, the opposite half of the same leaf was analyzed using rapid light curves (RLC) obtained through chlorophyll fluorescence analysis, aiming to quantify its ability to adapt to light stimuli (*Genty, Briantais & Baker, 1989*; *Fetene et al., 1997*). Leaves were subjected to increasing photon flux density along eight steps of 10 s each to obtain rapid light curves. RLCs were constructed to relate the apparent electron transport rate (ETR) and Genty's Yield parameter ($\Phi$ PSII = (Fm' − F) (Fm')$^{-1}$, where Fm' is the maximum fluorescence yield and F is the steady-state fluorescence yield (*Genty, Briantais & Baker, 1989*) under ambient light. The ETR was calculated as the product of quantum efficiency of PSII ($\Phi$ PSII) and the absorbed photosynthetically active radiation (PPFD) following the equation (*White & Critchley, 1999*): ETR = $\Phi$ PSII

$\times$ PPFD $\times$ 0.5 $\times$ 0.84. The constant 0.5 assumes an equal distribution between the two photosystems, and the ETR factor applied represents 84% of PAR absorbance. The maximum ETR value ($ETR_{max}$) was determined following (*Potvin, Lechowicz & Tardif, 1991*). After photosynthesis and chlorophyll fluorescence determinations, each leaf was sampled to determine the nitrogen concentration on a mass (N as % of leaf dry weight) basis using standard methods (*Kjeldahl, 1883*).

## Leaf morphophysiology and anatomy

For anatomical analysis, 0.5 $\times$ 0.02 cm leaf samples were fixed in 2.5% glutaraldehyde and 0.1 M phosphate buffer (*Gabriel, 1982*). The samples were dehydrated in an ethyl alcohol series, embedded in hydroxymethyl methacrylate (*Gerrits & Smid, 1983*), sectioned and stained with 0.5% toluidine blue O (*O'Brien, Feder & McCully, 1964*). Anatomical sections were photographed using a BX-50 microscope (Olympus, Westborough, MA, USA), and parameters were measured using Image Pro-Plus software (Media Cybernetics, Rockville, MD, USA). The parameters leaf (LT) and mesophyll thickness (MT) and the ratio of palisade/spongy thickness (PS) were first determined. The cross-sectional area of the mesophyll occupied by intercellular spaces (IS) was quantified in a 200 $\mu$m wide section (*Nobel, 2005*).

Stomatal and vein densities were determined by observing the abaxial surface of leaf samples previously clarified with 5% NaOH and 5% NaClO (*Shobe & Lersten, 1967*) and stained with aqueous safranin. The stomatal density (SD) was estimated by determining the number of stomata in an area of 0.37 mm$^2$ using a 20$\times$ objective from a BX-50 microscope (Olympus, USA). Vein density (VD) was estimated by determining the total length of the venation over a leaf area of 9.29 mm$^2$ (*Carins Murphy, Jordan & Brodribb, 2012*) using a 4$\times$ objective on a BX-50 microscope (Olympus, USA). Both stomatal and vein density quantifications were determined using Image Pro-Plus software (Media Cybernetics).

The morphophysiological parameters of leaf area (LA), leaf succulence (LS) and specific leaf area (SLA) were determined. LA was quantified *via* digital photography using Image Pro-Plus software (Media Cybernetics), while LS and SLA were, respectively quantified by the ratios '(saturated fresh weight-dry weight)/area' and 'area/dry weight' following standard procedures (*Mantovani, 1999*; *Mantovani, Pereira & Mantuano, 2017*). The anatomical and morphophysiological parameters cited above were determined for the same leaves used to evaluate photosynthetic parameters.

## Plant growth rate

The influence of leaf enlargement and photosynthesis on the final growth of *E. aureum* was investigated by determining relative growth rate (RGR, day$^{-1}$) and net assimilation rate (NAR, g cm$^{-2}$ day$^{-1}$) parameters (*Poorter et al., 2009*). For this, dry mass and leaf area data published by *Brito et al. (2022)*, quantified for the same individuals and treatments investigated here, were used. The total plant dry mass ($TDM_i$) and total leaf area ($LA_i$) for 20 plant cuttings with four expanded leaves were quantified to represent the initial of the experiment (*Brito et al., 2022*). After 10 months of growth, the parameters final total plant dry mass ($TDM_f$) and final total leaf area ($LA_f$) were quantified for the 20 individuals here

studied, marking the end of experiment. Relative growth rate (RGR) and net assimilation rate (NAR) were calculated following *Hunt et al. (2002)*: $RGR = (\ln TDM_f - \ln TDM_i/T)$ and $NAR = ((TDM_f - TDM_i)/T) \times (\ln LA_f - \ln LA_i)/(LA_f - LA_i))$, where T represents the time interval of 300 days between the initial and end of experiment.

## Statistical analysis

Data were compared pairwise for light (low light *versus* high light) and for growth direction (horizontal *vs.* climbing) conditions. Data for the same growth direction (horizontal and climbing) were compared under different light intensities (low light *vs.* high light) and vice versa. First, normality was evaluated using the Shapiro–Wilk test. Accordingly, when necessary a t test was used for parametric and Mann–Whitney test for nonparametric data. Statistical analyses were obtained using software R (*R Core Team 2021*) (Rx64, 3.4.1) with the significance of $P < 0.05$ (*Zar, 1996*).

# RESULTS

## Photosynthetic parameters

There were significant differences in photosynthetic parameters between plants of *E. aureum* under different light intensities. Plants under high light (HL-horiz and HL-climb) presented 2–3 times higher maximum electron transport rates ($ETR_{max}$) than plants under low light (LL-horiz and LL-climb), reaching values around 130 $\mu$mol m$^{-2}$ s$^{-1}$ under high light (Fig. 2, Table S1). In contrast, there was no difference in the $ETR_{max}$ values for plants at the same light intensity. Although some of the higher values of $A_{max}$ and LSP were found for HL-climb plants (Table S1), both parameters were statistically similar in all treatments, regardless of growth or light conditions (Fig. 2). The $A_{100}$ value for LL-horiz was significantly lower ($P = 0.01$) than $A_{1000}$ for HL-climb treatments, respectively $1.57 \pm 0.81$ and $5.3 \pm 2.4$ $\mu$mol m$^{-2}$ s$^{-1}$ (Fig. S1). Comparing with field data, the similar LSP level of PPFD here obtained for both LL-horiz and HL-climb treatments was achieved only at canopies but not at understory, under natural conditions. The Rd was more than twice higher under high light in comparison to the low light treatment, while HL-climb plants presented Rd values significantly higher than all other treatments (Fig. 2).

## Leaf morphophysiology and anatomy

Leaf areas produced at the apex of HL-climb plants varied from 900 to more than 1,370 cm$^2$, 9–13 times larger than in all other treatments, including LL-climb (Table 1; Table S1). Plants grown under horizontal growth direction presented leaf area of 60 and 115 cm$^2$ for low and high light conditions, respectively. Leaves produced under high light conditions presented two times lower SLA values in comparison to leaves developed under low light conditions. Leaf succulence varied from 224 to 337 g m$^{-2}$, being slightly higher for HL-horiz and HL-climb (Table 1). Leaf nitrogen concentration (% of leaf dry weight) was always around 3% in all treatments (Table 1). All *E. aureum* leaves presented a dorsiventral structure. Leaf thickness was always around 300–330 $\mu$m (Table 1), except for the leaves from HL-horiz where values reached 398 $\mu$m. Under highlight treatments the mesophyll thickness was higher for both, horizontal and climbing growth. Both the palisade/spongy

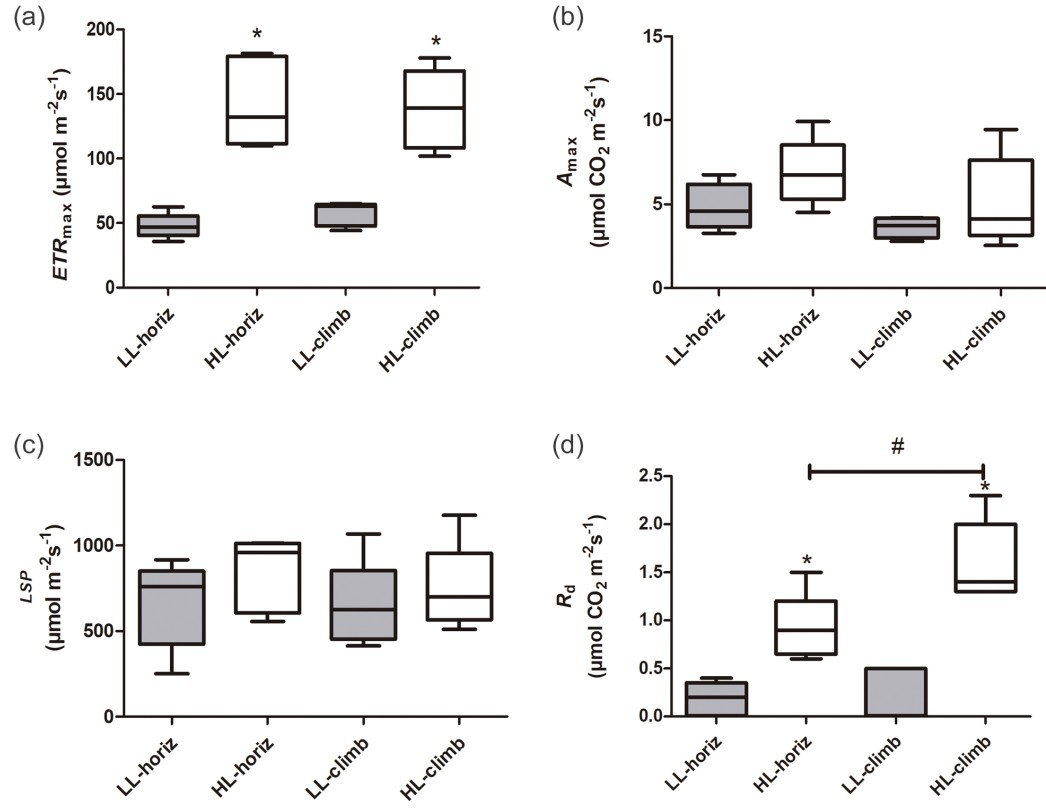

**Figure 2 Photosynthetic parameters of *Epipremnum aureum* leaves under different light conditions (low and high light) and growth directions (horizontal and vertical).** (A) Maximum electron transport rate (ETR$_{max}$); (B) maximum carbon assimilation rate (A$_{max}$); (C) light saturation point (LSP); and (D) dark respiration rate (Rd). An asterisk (*) indicates statistical differences between different light conditions; a number sign (#) indicates statistical differences between different growth directions ($p < 0.05$; $n = 5$). LL-horiz: low light horizontal; HL-horiz: high light horizontal; LL-climb: low light vertical; and HL-climb: high light vertical.

thickness ratio and the proportion of cross-sectional area occupied by intercellular spaces were similar along all treatments (Table 1). HL-climb had almost two times higher stomatal density than LL-horiz, while for vein density HL-climb had 1.75 times higher values than LL-horiz (Fig. 3).

## Plant growth rate

Individuals of *E. aureum* in all treatments showed substantial growth after 10 months of the experiment (Fig. S2). For the same horizontal or vertical orientation, higher relative growth rates (RGR) were exhibited by plants in high-light treatments (Table 1). The highest mean RGR value among treatments was 0.021 mg g$^{-1}$ day$^{-1}$ for HL-climb individuals (Table 1), while the same parameter was around 0.012 mg g$^{-1}$ day$^{-1}$ for individuals under low-light treatments. A similar pattern was observed for the net assimilation rate (NAR), where, for the same orientation, individuals in high-light treatments exhibited higher NAR

**Table 1** **Leaf morpho-physiology and anatomy of *Epipremnum aureum* growing under different light conditions (low and high light) and growth directions (horizontal and vertical).** Capital letters compare low light *vs.* high light in the same growth (horizontal or climbing) direction. Lowercase letters compare horizontal *vs.* climbing in the same light condition (low light or high light) ($p < 0.05$; $n = 5$).

| | | LL-horiz | HL-horiz | LL-climb | HL-climb |
|---|---|---|---|---|---|
| *Morpho-physiology* | Leaf area (cm$^2$) | 88.91 ± 30.28 Aa | 129.88 ± 69.69 Aa | 112.20 ± 37.09 Aa | 1,161.96 ± 160.10 Bb |
| | Leaf succulence (g m$^{-2}$) | 298.40 ± 28.91 Aa | 337.80 ± 35.14 Aa | 224.20 ± 19.84 Ab | 252.20 ± 22.25 Ab |
| | Specific leaf area (cm$^2$ g$^{-1}$) | 311.90 ± 62.09 Aa | 155.19 ± 27.96 Ba | 340.14 ± 31.62 Aa | 156.04 ± 7.44 Ba |
| | Nitrogen (%) | 3.33 ± 0.47 Aa | 3.47 ± 0.48 Aa | 3.27 ± 0.33 Aa | 3.27 ± 0.33 Aa |
| *Anatomy* | Leaf lamina thickness (μm) | 303.80 ± 35.34 Aa | 398.29 ± 45.52 Ba | 314.20 ± 22.50 Aa | 332.60 ± 30.75 Ab |
| | Mesophyll thickness (μm) | 204.00 ± 27.54 Aa | 310.40 ± 37.45 Ba | 209.20 ± 14.92 Aa | 274.60 ± 28.79 Ba |
| | Palisade/Spongy ratio | 0.51 ± 0.11 Aa | 0.54 ± 0.11 Aa | 0.64 ± 0.17 Aa | 0.51 ± 0.02 Aa |
| | Intercellular space per mesophyll area (%) | 20.68 ± 6.36 Aa | 19.90 ± 3.93 Aa | 23.23 ± 3.00 Aa | 19.70 ± 5.31 Aa |
| | Vein density (cm cm$^2$) | 30.71 ± 6.52 Aa | 37.79 ± 3.58 Aa | 24.46 ± 3.99 Aa | 42.74 ± 12.61 Ba |
| | Stomatal density (mm$^{-2}$) | 36.44 ± 14.85 Aa | 45.55 ± 8.05 Aa | 34.17 ± 11.39 Aa | 61.50 ± 6.24 Bb |
| *Plant growth rate parameters* | RGR (day$^{-1}$) | 0.013 ± 0.001 Aa | 0.018 ± 0.001 Ba | 0.011 ± 0.001 Aa | 0.021 ± 0.001 Bb |
| | NAR (g cm$^{-2}$ day$^{-1}$) | 0.187 ± 0.043 Aa | 0.422 ± 0.121 Ba | 0.126 ± 0.029 Aa | 0.725 ± 0.331 Ba |

values. HL-climb individuals exhibited four to five times higher NAR values compared to LL-horiz and LL-vertical individuals (Fig. S3).

## DISCUSSION

The present study investigates the potential effects of leaf enlargement on the overall performance and maintenance of the aroid vine *Epipremnum aureum* on canopies. Maximum photosynthesis obtained *via* light curves is apparently not able to compensate for the higher production and maintenance costs of the *E. aureum* large leaves in our experiment. Although presenting some of the maximum values of A$_{max}$ and LSP found in this experiment in high light treatments, there was no significant increase of these parameters when larger leaves were statistically compared to smaller ones. However, one point emerges from the photosynthetic light-curves here obtained: compared with field observations, the high light levels able to maximize photosynthesis occur only at canopies where larger leaves of *E. aureum* are found, not at understory where its smaller leaves are maintained (*Mantovani, Pereira & Mantuano, 2017*). Additionally, a considerable increase in respiration rate was detected for the large leaves of HL-climb in comparison to all treatments, including HL-horiz. An increase in leaf area can contribute to a greater light-harvesting capacity (*Van Ieperen, 2012*). However, it does not directly represent a proportional improvement in carbon acquisition, as large leaves require functional adjustments demanded by size enlargement. Examples are a higher resistance to mechanical bending, evapotranspiration, augmented temperatures and herbivory (*Niklas, 1994*; *Kozlov, Zverev & Zvereva, 2022*) which increase costs of production and maintenance per unit of leaf area (*Niklas, 1999*; *James & Bell, 2000*). Results obtained here indicate a

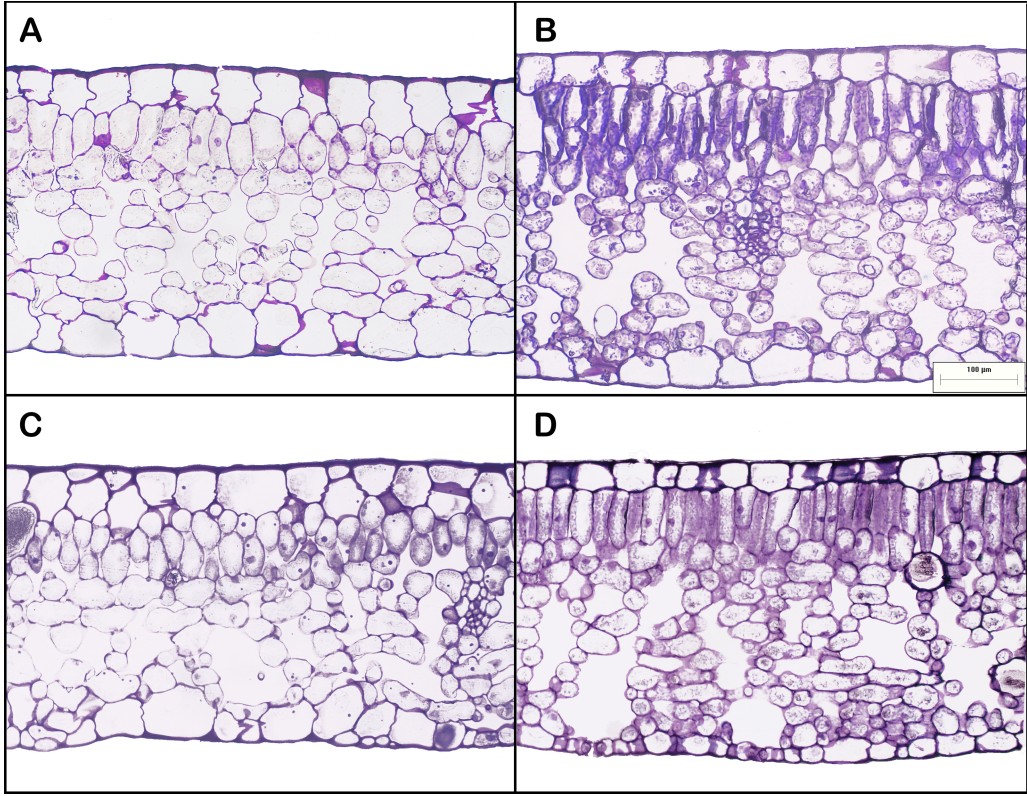

**Figure 3** **Leaf anatomy of *Epipremnum aureum* leaves under different light conditions (low and high light) and growth directions (horizontal and vertical).** (A) Low light horizontal (LL-horiz); (B) high light horizontal (HL-horiz); (C) low light climbing (LL-climb); and (D) high light climbing (HL-climb). Scale bar = 100 μm for all images.

higher functional and structural cost per unit area of large leaves that is not counterbalanced at least in by its respective maximum photosynthesis in this experiment in short term.

In the long term, an extended lifespan can improve investment made in large leaves by increasing its lifetime for carbon uptake (*Wright et al., 2004*; *Kikuzawa & Lechowicz, 2006*). Previous works compared the leaf longevity in five climbing aroid vines and four co-occurring terrestrial aroid herbs (*Shiodera, Rahajoe & Kohyama, 2008*). The leaves of aroid vines presented higher lifespan from 28 to 40 months and a two times lower SLA while leaves of the co-occurring terrestrial aroids lasted only from 3 to 17 months presenting higher SLA. Combining data for both terrestrial and aroid vine leaves revealed a significant decreasing relationship ($R^2 = 0.95$; $P < 0.001$, Appendix 1 from *Shiodera, Rahajoe & Kohyama, 2008*) among respective SLA and lifespan values. The decreasing SLA pattern is a potential predictor of an increasing leaf lifespan (*Reich, Walters & Ellsworth, 1992*). Larger leaves of *E. aureum* here had 1.5 times lower SLA values in comparison to smaller ones under shade, stimulating the idea that that the return of investments made in large leaves of *E. aureum* could be paid back in the long term (*Wright et al., 2004*; *Wang et*

*al., 2023*; *Castorena et al., 2022*), *via* increased lifespan. More investigations are necessary to confirm this hypothesis.

Regardless of the cost of large leaves, HL-climb individuals exhibited the highest relative growth rate (RGR) and net assimilation rate (NAR) values compared to other treatments, showing an improvement in plant growth with increasing leaf size. This improvement is also observed in other allo- and heteromorphic aroid vines, which often show an increase in total root and shoot size and dry mass associated with leaf enlargement (*Filartiga, Vieira & Mantovani, 2014*; *Brito et al., 2022*). In the literature, RGR values are better correlated with NAR than with SLA values under more exposed conditions (*Shipley, 2002*; *Shipley, 2006*), which also occurred here: while RGR and NAR values increased for high-light plants of *E. aureum*, SLA values simultaneously decreased. Considering that NAR is a physiological measure of whole-plant net photosynthesis and that SLA is a morphological component determined by leaf density and thickness (*Shipley, 2002*), the inverted pattern found here between both parameters suggests a trade-off between photosynthesis and anatomy (*Poorter et al., 2009*), mediated by the increase in leaf area size.

Not only *E. aureum* leaves but also its roots, stem and consequently, whole plant dry masses increased nine times for HL-climb in comparison to LL-horiz individuals after 10 months of experiment (*Brito et al., 2022*). According to literature (*Wright et al., 2017*), there are two main hypotheses for selective advantages of plants investing in a larger leaf area: (1) the investment in leaf area is not proportionally offset by the investment in branches, resulting in a saving in the production of branches through the production of larger leaves even in herbs; or (2) the increased thickening of the boundary layer, which allows leaves to reach or maintain optimal temperatures for photosynthesis when the temperature outside the leaf is unfavorable in the early mornings. Both hypotheses seem applicable to the results found here for the aroid vine *E. aureum*.

First the climbing habit of the aroid vines is mechanically supported by its hosts. This precludes an elevated investment in mechanical supportive tissues in stems (*López-Portillo et al., 2000*), which is commonly dependent on a large carbon allocation to thick and lignified structures (*e.g.*, sclerenchyma and vascular bundles) (*Niklas, 1994*), even in monocotyledon stems (*Masselter et al., 2016*; *Hesse, Wagner & Neinhuis, 2016*). This would result in saving more carbon to be invested in larger leaves. Second, photosynthesis in aroid vines could be improved by the increase of temperature that usually occurs when large leaves are exposed to higher light conditions (*Li et al., 2013*). Large leaves of the aroid vines are maintained at canopy microsites where air temperatures can reach 35 °C (*Filartiga, Vieira & Mantovani, 2014*; *Mantovani, Brito & Mantuano, 2018*) which is close to the optimum temperature condition for photosynthesis at the beginning of the morning (*Falster & Westoby, 2003*). Although the LSP value inducing maximum photosynthesis in our experiment was similar for larger and smaller *E. aureum* leaves, it did not reflect the respective light environment experienced by both leaves under natural conditions. The PPFD values experienced by large *E. aureum* leaves under natural canopy conditions (*Mantovani, Pereira & Mantuano, 2017*) are similar to the LSP values around 1.000 $\mu$mol m$^{-2}$s$^{-1}$ obtained here in this experiment *via* gas exchange curves. However, this elevated PPFD value is never attained where the smaller *E. aureum* leaves occur, as maximum PPFD

is around 100 $\mu$mol m$^{-2}$s$^{-1}$ in the understory (*Mantovani, Mantuano & De Mattos, 2017*) (Fig. S1). As such, only the large leaves of *E. aureum* positioned in the canopies would experience higher photosynthetic rates at LSP under natural condition. While previous field work showed higher photosynthetic rates for large canopy leaves of the aroid vine *R. oblongata* compared to smaller ones produced in the understorey (*Mantuano et al., 2021*), field investigations are necessary to confirm this hypothesis for *E. aureum*. Finally, the high Amax and LSP values found here under experimental conditions for the smaller leaves of the aroid vine *E. aureum* could be related to its effective invasive capacity around the world, taking advantages to occupy illuminated disturbed areas and forest edges as an alien species (*Moodley, Proches & Wilson, 2017*; *Mcalpine, Jesson & Kubien, 2008*).

As expected, leaf dark respiration rates were higher for HL in comparison to LL individual plants, reflecting the higher metabolic cost of protein turnover and phloem-loading of photosynthates under more illuminated conditions (*Cannell & Thornley, 2000*). Although with plenty of irrigation (*Brito et al., 2022*) and no chlorosis at all in *E. aureum* leaves during our experiment, the ETR$_{max}$ were higher under high light conditions. The increase of these rates are usually related to thermo and photoprotection capacities (*Demmig-Adams et al., 1989*; *Ribeiro et al., 2009*; *Slot & Winter, 2018*). The same pattern was obtained under field conditions for *E. aureum* when higher ETR values occurred for large leaves under higher light conditions at canopy in comparison to smaller leaves positioned understory (*Mantovani, Pereira & Mantuano, 2017*).

In allo- and heteromorphic aroid vines, not only its petioles and leaf lamina but also its aerial roots increase in number, size and mass while ascending to the canopy (*Filartiga, Vieira & Mantovani, 2014*; *Brito et al., 2022*). The widening of aroid vine leaves is followed by the widening of root and stem xylem vessels (*Filartiga, Vieira & Mantovani, 2018*). This acropetal taper occurs in an opposite pattern to the usually found basipetal taper of trees (*López-Portillo et al., 2000*) and confers an efficient axial hydraulic conductivity to the aroid vine while climbing its hosts (*Filartiga, Vieira & Mantovani, 2014*; *Filartiga, Vieira & Mantovani, 2018*), including *E. aureum* (*Brito et al., 2022*). However, there are reports of an increased water use efficiency for large canopy leaves (*Mantuano et al., 2021*), indicating that an efficient root+stem hydraulic capacity could be decoupled from leaf transpiration for the aroid vines.

In fact, the vein (VD) and stomatal (SD) densities for the large leaves of *E. aureum* are only four mm mm$^{-2}$ and 60 mm$^{-2}$ respectively, while the low stomatal conductance for the same leaves is around 50 mmol H$_2$O m$^{-2}$s$^{-1}$ (*Mantovani, Pereira & Mantuano, 2017*). The same pattern is cited for another aroid vine *R. oblongata* (*Mantovani, Mantuano & De Mattos, 2017*). In the literature, VD values from 6 to more than 12 mm mm$^{-2}$ are reported in liana leaves (*Bai et al., 2024*), reaching more than 20 mm mm$^{-2}$ in angiosperms (*Sack & Scoffoni, 2013*; *Sack et al., 2013*). Stomatal density in liana leaves can reach more than 500 stomata mm$^{-2}$ (*Tay & Furukawa, 2008*). Finally large canopy leaves of aroid vines present a much higher leaf epidermal resistance to water loss than respective understory ones (*Mantovani, 1999*; *Mantovani, Pereira & Mantuano, 2017*). These data reinforce the hypothesis cited in the beginning of the discussion as an improvement in water balance

can positively influence leaf lifespans (*Wright, Reich & Westoby, 2001*; *Poorter, Bongers & Bongers, 2006*).

Surprisingly, while SLA values decreased under high light conditions, the leaf anatomy of *E. aureum* remained quantitatively conservative when comparing low and high light treatments in our experiment. A decrease in SLA values, while leaf thickness is maintained, is commonly associated with higher leaf densities (*Poorter et al., 2009*) probably induced here by the increased vein density (*Mantovani, Brito & Mantuano, 2018*) found in illuminated *E. aureum* leaves. Under field conditions, however, the larger and more illuminated leaves of the aroid vines growing in the canopy presented a thicker mesophyll with more intercellular spaces than the respective smaller ones produced in the understory (*Mantovani, Pereira & Mantuano, 2017*; *Mantovani, Mantuano & De Mattos, 2017*). Previous studies correlate the higher $A_{max}$ value found for large in comparison to smaller leaves of the aroid vine *R. oblongata* to a higher proportion of chloroplasts facing larger intercellular spaces of the mesophyll found in large leaves, while both large and small leaves presented the same nitrogen concentration of around 2.6% (*Mantuano et al., 2021*). Mesophyll conductance, biochemical content for the Calvin cycle (*e.g.*, N allocation to Rubisco (*Luo et al., 2021*)) and stomatal conductance are the main limiting factors for leaf photosynthesis when light, $CO_2$ and water conditions are optimal (*Grassi & Magnani, 2005*; *Gago et al., 2019*). The N concentration found here for *E. aureum* leaves was around 3% in all treatments, while stomatal density was always around 40–60 stomata $mm^{-2}$. In this sense, the similar intercellular space ratio correlates with the statistically similar $A_{max}$ values observed here for small and large leaves of *E. aureum*, reinforcing the idea that mesophyll conductance may influence aroid vine photosynthesis at leaf level.

Light intensity can change the size and anatomy of the aroid leaves (*Sims & Pearcy, 1992*; *Lorenzo, Mantuano & Mantovani, 2010*). Following (*Filartiga, Vieira & Mantovani, 2014*; *Brito et al., 2022*), the maximum leaf area (up to 1,000 $cm^2$) and light condition (650 $\mu$mol $m^{-2}s^{-1}$) found here for HL-climbing plants of *E. aureum* under experimental conditions were half of the values found under natural canopy conditions. The same difference applies to anatomy: the thickness of large leaves of *E. aureum* surpassed more than 500 $\mu$m under field conditions while here the mean thickness was not higher than 330 $\mu$m for large leaves. Results shown above indicate that size, morpho-physiology, anatomy and photosynthesis can be simultaneously modulated at the leaf level while different abiotic conditions occur for the aroid vine *E. aureum*. This modulation can improve functional resource allocation along the ascendant path from the understory toward the canopy of the aroid vines.

## CONCLUSIONS

Under experimental greenhouse conditions, large leaves of the aroid vine *Epipremnum aureum* presented a lower specific leaf area and higher carbon loss *via* dark respiration but similar anatomy and maximum photosynthetic rate on an area basis when compared to smaller leaves. However, compared with data obtained under natural conditions the light saturation levels for photosynthesis is achieved only by large leaves in canopies and not by smaller leaves in the understory. The higher relative growth rates presented

by vertical plants in high light conditions confirm this result. At the same time, higher net assimilation rates correlated with leaves that increased in size but did not change in mesophyll anatomy, indicating that leaf size enhanced total net photosynthesis under the experimental conditions. These results suggest that photosynthesis can offset the associated cost of producing large leaves in canopies. The aroid vine *E. aureum* modulated size, anatomy, morphophysiology and photosynthesis to optimize carbon gain for the whole plant under different abiotic conditions, explaining the increase in total size and biomass found for aroid vines.

## ACKNOWLEDGEMENTS

We are grateful to Dr. Benjamin Steinitz, Dr. Francisco Lobo, Dr. Ângela Vitória and Tatiane Vieira for their very constructive criticism and help with the development of the experiment. We are grateful to Dr. Antônio de Andrade for helping with experimental greenhouse facility. We also would like to give thanks to three reviewers and one Editor for their improvements to the manuscript.

### Funding

Conselho Nacional de Desenvolvimento Científico e Tecnológico (CNPq, Brazil) and CAPES (Coordenação de Aperfeiçoamento de Pessoal de Nível Superior, Brazil) supported Carolina Brito via a Master's grant. The funders had no role in study design, data collection and analysis, decision to publish, or preparation of the manuscript.

### Grant Disclosures

The following grant information was disclosed by the authors:
Conselho Nacional de Desenvolvimento Científico e Tecnológico (CNPq, Brazil).
CAPES (Coordenação de Aperfeiçoamento de Pessoal de Nível Superior, Brazil).

### Competing Interests

The authors declare there are no competing interests.

### Author Contributions

- Carolina Brito conceived and designed the experiments, performed the experiments, analyzed the data, prepared figures and/or tables, and approved the final draft.
- Dulce Mantuano conceived and designed the experiments, analyzed the data, prepared figures and/or tables, and approved the final draft.
- Karen L.G. De Toni analyzed the data, prepared figures and/or tables, and approved the final draft.
- André Mantovani conceived and designed the experiments, performed the experiments, analyzed the data, prepared figures and/or tables, authored or reviewed drafts of the article, and approved the final draft.

## Data Availability

The raw measurements are available in the Supplementary File.

## Supplemental Information

Supplemental information for this article can be found online at http://dx.doi.org/10.7717/peerj.19214#supplemental-information.

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
