# Peer review of "Increasing leaf sizes of the vine Epipremnum aureum (Araceae): photosynthesis and respiration"

_PeerJ, doi:10.7717/peerj.19214_

## Round 0.1 · original submission · Major Revisions

All three reviewers have suggested major revisions. In particular, Reviewer 3 recommends inclusion of the theoretical basis of the concepts explored in the study. Among the detailed comments given, Reviewer 2 has raised concerns about the computation of the carbon balance. Reviewer 1 also queries the calculation of the carbon balance from instantaneous measurements of photosynthesis and respiration. All three reviewers have requested a more comprehensive characterization of the light environment.

In addition to revisions on the manuscript, please address all points raised by the three reviewers in a point-by-point description on a separate document.

Reviewer 1 ·

Basic reporting

Clear and unambiguous, professional English used throughout.
Yes
 Literature references, sufficient field background/context provided.
Yes
 The article should include sufficient introduction and background to demonstrate how the work fits into the broader field of knowledge. Relevant prior literature should be appropriately referenced.
Yes
 Professional article structure, figures, tables. Raw data shared.
Yes
 Self-contained with relevant results to hypotheses.
Not Enough

Experimental design

Original primary research within Scope of the journal.
Yes
 Research question well defined, relevant & meaningful. It is stated how the research fills an identified knowledge gap.
Yes
 Rigorous investigation performed to a high technical & ethical standard.
Yes
 Methods described with sufficient detail & information to replicate.
it needs to be revised

Validity of the findings

Impact and novelty is not assessed. Meaningful replication encouraged where rationale & benefit to literature is clearly stated.
OK
 All underlying data have been provided; they are robust, statistically sound, & controlled.
Not enough
 Conclusions are well stated, linked to original research question & limited to supporting results
Yes, but it needs to be revised

Additional comments

General comments

The authors investigated physiological and morphological change in leaves of a vine Aroid species, E. aureum, by changing both light environment and growth direction, try to evaluate the carbon balance especially for extremely large leaves of individuals in high-light and of vertical growth. The experimental condition of "high-light and climbing (HL-climb)" let the species make big leaves, but their Amax was not significantly changed compared to leaves under other conditions while Rd was significantly higher. From the results, the authors found that HL-climb leaves were not superior in carbon balance in the short term. However, the authors infer that the large plasticity in leaf traits suggests that this species optimizes whole-plant carbon balance.
The manuscript is clearly written, well organized and readable. The method of experimentally changing the light environment and growth direction to observe changes in the physiological and morphological characteristics of leaves is sophisticated and provide important data for understanding the species' plasticity. However, unfortunately, another important hypothesis alongside carbon acquisition ability per leaf area, the length of leaf lifespan (mentioned by the authors), has not been verified, so it is unclear whether the carbon balance of HL-climb is superior in supporting whole-plant growth.

 If you have data on leaf lifespan, please add it.

 Leaf carbon acquisition ability was evaluated using short-term or instantaneous measurements such as Amax and ETR, but I think the discussion would be more in-depth if you added an evaluation based on the change in whole-plant mass over the 10-month experimental period using like RGR, NAR.

 Add information on the light environment; data showing the light environment during experimental period, such as the daily average photon flux density and the shading rate. Amax and Rd, the values seem not to be very high even under the high-light conditions. Perhaps it is not growing in a very bright environment, or it is a species with characteristics of a late-successional species. Also, if there is any data on the natural light environment in which wild E. aureum grows, include it.


Other comments

Brito et al. (2022) revealed the conditions under which leaves of the species increase in size. Does current study use the data from that study or is it a new experiment?

Photos or diagrams of the experiment would help to understand.

L135 & L137: Literature citations are given numerically.

L310-311 “Following (Filartiga, Vieira & Mantovani, 2014; Brito et al., 2022), the maximum leaf area (up to 1000 cm2) and light condition (650 μmol m-2 s-1) found here for HL-climbing plants of E. aureum under experimental conditions were half of the values found under natural canopy conditions.”
Provide information about the general light environment where this species grows and the light environment during the experiment in the Materials and Methods.
Calculating carbon balance using traits of individuals grown under natural field conditions (higher-light conditions?) may lead to a different conclusion?

L328-331 “Compared with data obtained under natural field conditions, the results showed that the aroid vine E. aureum modulated size, anatomy, morphophysiology and photosynthesis to optimize carbon gain for the whole plant under different abiotic conditions, explaining the increase in total size and biomass found for aroid vines in the canopy”
The discussion is generally thorough and ranging, and the data indeed indicate physiological and morphological plasticity in species. But it would be a stretch to say that this explains that the species optimize whole-plant carbon gain.

Table 1
L3 “Leaves from HL-climb plants are 9-13 times larger than leaves from the other treatments.”
Remove the sentence, not needed.

Reviewer 2 ·

Basic reporting

Format of the paper and the presentation are agreeable.
Figure 1 and 2: legends are missing.
Fig 2: labelling is not done and it needs labelling.

Experimental design

The experimental design used here is a 2×2 factorial design. However, authors have presented only the main effects of light and growth direction. Is there any interaction between these two factors?

Need to re-think about the number of replicates per treatment, only five replicates per treatment, and also measurements were taken from only one leaf at the apex, only one level and only once.

Validity of the findings

• Introduction is not well focused towards the objectives of the study or the plant carbon balance. Its vague.
• Methodology section needs lot of improvements: The information on experimental conditions in the green house, eg; light intensities of LL and HL, its diurnal variation, variation over 10 months, whether irrigated / not, whether fertilizer applied / not), are not given. The previous paper referred to methodology (Brito et al., 2022) does not give experimental conditions.
• The title indicates Carbon Balance, but in the methods section, there is no any description on how C-balance is calculated.
• From the discussion, it appears that authors assume C-balance as the difference between Photosynthesis and Respiration. Both these parameters have been derived from light response curves, therefore fix values. However, both photosynthesis and respiration of plants show diurnal variation due to the effects of temperature and other contributing factors such as VPD. Furthermore, the light response has been measured using only one leaf of the canopy (at the apex level) but there is a variation of light response in different canopy levels specially with vertical climbing treatments (LL and HL Climb). Those are not considered.
• Hence, calculating C-balance using constant photosynthesis and respiration is not correct and it should be integrated at a given time scale (i.e. hourly/daily etc.). Additionally, not only photosynthesis and respiration, but also other processes such as C-allocation to tissues, non-structural C storage also affect C-balance. However, this study does not consider such processes therefore C-balance concept used in this study is not reliable.
• In the methodology section, absorbed PPFD values are given, the authors have not specified the incident PPFD and how they calculated absorbed PPFD fraction (line 109).
• According to Lines 105-106, Lines 127-129 and Lines 153-155, they have used one completely expanded leaf which was used for measuring photosynthesis light response, to measure leaf N, morpho-physiological parameters and anatomical parameters. Is one leaf adequate for all analyses ? is it correct, again theses parameters can vary with the canopy level of the LL and HL climbing treatments. So one leaf near the apex will not explain the plant level mechanism.

• The study does not consider all processes required to explain carbon balance of a plant. Therefore C-balance concept used in this study is not reliable.
• Line 197-198: Photosynthesis is apparently not able to compensate for the higher production and maintenance costs of the E. aureum large leaves in our experiment . This is not a valid statement as they have not separately measured growth and maintenance respiration.

Additional comments

All covered above.

·

Basic reporting

Dear authors, I have read with great interest your study. In general, I think it is a good one, but it would increase its value if it were framed in a more general context. And although you didn’t measure leaf longevity, a key parameter that could explain the phenomenon, you recognized its importance at the end of the study, which is somehow disappointing. Before going to Gmax, there is another botanical phenomenon that came to mind while reading your manuscript; this is heteroblasty, or the abrupt change in leaf form and function with plant development. The closest example of heteroblasty for me is eucalypts with their juvenile and mature leaves. I could say that your species system is heteroblasty, because according to what I read, yours is a change in development. If you agree with me in that your species is heteroblasty, this would be a good frame where to introduce your study. But, most importantly, I think you should read Castorena’s article about the Gmax theory (Castorena et al., 2022), which, along with metabolic scaling theory (MST), is based on the works of Kikuzawa on leaf longevity (citations below). With all this, what I mean is that your study would benefit quite a bit from more theory from which you could set hypotheses or expectations. In the current form, your study is rather exploratory, which is not bad, but it could be better.

In the experimental setting, it is not clear whether the light environment at the canopy, which experimentally is denoted as “high light”, is total open canopy or it’s just lighter than the “low light” situation. I mean, these plants in the forest may never attain full access to light.
You need to include a figure with the experimental setting.
I hope my points will be of help when preparing the new version of your study.

Alex Fajardo.-


Minor comments:
L101, … following?
L110, I think a new figure 1 depicting the experimental setting either as a diagram or actual photographs would help to visualize the experiment.
L112, you need to correct the link to references; here, it should be “…see Brito et al. (2022)”.
L119, I don’t understand here why you mention 5 individuals. From above, it’s clear the experiment is a 2x2 factorial, which means 4 treatments, with 10 individuals each. Are there blocks?
127-8, citations are not properly referenced. I won’t bring this issue again.
L168, I think one parameter you miss in your study is leaf longevity, which is key in the carbon gain formula you are trying to convey.
L171-2, I don’t understand why you set your data analysis explanation in this way. This is just a normal ANOVA, where you first will assess the significance of both factors (light and position) and see if there is any interaction. If at least one of the factors proves to be significant (assuming an alpha value of 0.05), then you run post-hoc analysis to see differences among the 4 treatments.
L175, why Mann-Whitney test for nonparametric data? How was the error distribution of your data? Knowing the nature of your data, I would expect a normal error distribution, therefore the analysis should be parametric.
L192-5, it would be advisable to add standard error values to the means reported.
L197, here you mention leaf N concentrations, but I am sure you never introduced (and how you measured it) this trait in the methods.
L210-1, for this conclusive statement to be true you need to consider leaf longevity; there is no other way. See the works of Kikuzawa and his collaborators (e.g., Kikuzawa, 1991, Kikuzawa and Ackerly, 1999, Kikuzawa and Lechowicz, 2006), and the most recent article in this respect (Castorena et al., 2022), which boldly states that all plants have a similar lifetime net carbon gain per body mass. Following the Gmax theory (Castorena et al., 2022), I could say that
L227-9, this comparison makes sense to me; longer longevity goes along with lower SLA (or higher LMA), the opposite being true.
L233, correct!
L234, you need to read the list of studies I provided to see there is much written about this.
L245-9, correct.



References
Castorena M, Olson ME, Enquist BJ, Fajardo A. 2022. Toward a general theory of plant carbon economics. Trends in Ecology and Evolution, 37: 829-837.
Kikuzawa K. 1991. A cost-benefit analysis of leaf habit and leaf longevity of trees and their geographical pattern. The American Naturalist, 138: 1250-1263.
Kikuzawa K, Ackerly D. 1999. Significance of leaf longevity in plants. Plant Species Biology, 14: 39-45.
Kikuzawa K, Lechowicz MJ. 2006. Toward a synthesis of relationships among leaf longevity, instantaneous photosynthetic rate, lifetime leaf carbon gain, and the gross primary production of forests. The American Naturalist, 168: 373-383.

Experimental design

Correct (see above).

Validity of the findings

Correct (see above).

Additional comments

No additional comments (see above).

---

## Round 0.2 · Minor Revisions

While the authors have addressed most of the issues raised by the reviewers, Reviewer 1 highlights a few more outstanding issues that have to be addressed. In particular, essential data and analysis are not provided to clarify the cost and benefit in carbon balance of producing large leaves.

Reviewer 1 has given further specific comments and instructions to address the outstanding issues.

It is clear that Reviewer 1 requires addressing of these specific issues before recommending acceptance. Therefore, the authors are advised to pay particular attention to address all the issues that have been highlighted by Reviewer 1.

Reviewer 1 ·

Basic reporting

good

Experimental design

good

Validity of the findings

To present the fact that the physiological and morphological characteristics of E. aureum leaves differ depending on the light environment and growth style, there are no major problems with the revised manuscript, and the discussion based on the literature is enough. On the other hand, in terms of clarifying the cost and benefit in carbon balance of producing large leaves, the essential data and analysis are not provided.

I agree that it is difficult to get information of leaf lifespan of the species. For reference, I think it will be possible to discuss the cost/gain balance of a leaf and the minimum leaf lifespan required to compensate its cost, if the authors calculate the net assimilation rate (NAR) and compare it with leaf mass per area by using data on light curves, PPFD, dry mass change over a 10-month period, leaf area, etc.

Additional comments

L32-35 in revised MS: “…the high light levels able to saturate photosynthesis under field conditions are achieved only by larger leaves of E. aureum positioned at canopies (PPFD around 1000 µmol m-1 s-1), not occurring at understory where smaller leaves are positioned (PPFD around 100 µmol m-1 s-1).”
If calculation was provided for the difference in the amount of photosynthesis per leaf area per time among the treatments of this experiment, it would be more persuasive.

L378-381 in revised MS: “The results showed that the aroid vine E. aureum modulated size, anatomy, morphophysiology and photosynthesis to optimize carbon gain for the whole plant under different abiotic conditions, explaining the increase in total size and biomass found for aroid vines in the canopy.”
This is not shown in this study... to prove that the carbon balance is optimized, actual data from the experiment, not just cited references, is needed such as leaf life span, the cumulative amount of photosynthesis, allocation rate of assimilates to leaves, and so on.

Reviewer 2 ·

Basic reporting

The author has addressed and attended most of the suggestions and agree with the revised version.

Experimental design

The author has addressed and attended the suggestions and agree with the revised version.

Validity of the findings

The author has addressed and attended most of the suggestions and agree with the revised version.

Additional comments

None.

---

## Round 0.3 · accepted · Accept

The authors have addressed all comments of the reviewers, given via two rounds of review, satisfactorily. Therefore, the present version of the manuscript is ready for publication.